# Structural, Electronic and Magnetic Properties of $Mn_2Co_{1-x}V_xZ$ (Z = Ga, Al) Heusler Alloys: An Insight from DFT Study

**Fatima Abuova [1,\*], Talgat Inerbaev [1,2], Aisulu Abuova [1], Nurpeis Merali [1], Nurgul Soltanbek [1], Gulbanu Kaptagay [3], Marina Seredina [4] and Vladimir Khovaylo [4]**

[1] Department of Technical Physics, International Department of Nuclear Physics, New Materials and Technologies, L.N.Gumilyov Eurasian National University, Nur-Sultan 010000, Kazakhstan; talgat.inerbaev@gmail.com (T.I.); aisulu-us1980@yandex.kz (A.A.); nurpeis.93@mail.ru (N.M.); sns.nugul@mail.ru (N.S.)

[2] Laboratory of Phase Transitions and Diagrams of the Earth's Matter under High Pressure, Sobolev Institute of Geology and Mineralogy, 630058 Novosibirsk, Russia

[3] Physics Department, Kazakh National Women's Teacher Training University, Almaty 050000, Kazakhstan; gulbanu.kaptagai@mail.ru

[4] Department of Functional Nanosystems and High-Temperature Materials, National University of Science and Technology "MISIS", 119049 Moscow, Russia; nmseredina@gmail.com (M.S.); khovaylo@misis.ru (V.K.)

\* Correspondence: Fatika_82@mail.ru; Tel.: +7-778-647-8343

**Abstract:** Structural, electronic, and magnetic properties of $Mn_2Co_{1-x}V_xZ$ (Z = Ga, Al, x = 0, 0.25, 0.5, 0.75, 1) Heusler alloys were theoretically investigated for the case of $L2_1$ (space group $Fm\bar{3}m$), $L2_{1b}$ ($L2_1$ structure with partial disordering between Co and Mn atoms) and XA (space group $F\bar{4}3m$) structures. It was found that the XA structure is more stable at low V concentrations, while the $L2_1$ structure is energetically favorable at high V concentrations. A transition from $L2_1$ to XA ordering occurs near x = 0.5, which qualitatively agrees with the experimental results. Comparison of the energies of the $L2_{1b}$ and XA structures leads to the fact that the phase transition between these structures occurs at x = 0.25, which is in excellent agreement with the experimental data. The lattice parameters linearly change as x grows. For the $L2_1$ structure, a slight decrease in the lattice constant *a* was observed, while for the XA structure, an increase in *a* was found. The experimentally observed nonlinear behavior of the lattice parameters with a change in the V content is most likely a manifestation of the presence of a mixture of phases. Almost complete compensation of the magnetic moment was achieved for the $Mn_2Co_{1-x}V_xZ$ alloy (Z = Ga, Al) at x = 0.5 for XA ordering. In the case of the $L2_1$ ordering, it is necessary to consider a partial disorder of atoms in the Mn and Co sublattices in order to achieve compensation of the magnetic moment.

**Keywords:** Heusler alloys; compensated ferrimagnetism; DFT; structural ordering; electronic structure

## 1. Introduction

In recent years, Heusler alloys have been studied extensively because of their diverse magnetic phenomena [1]. Among them, Mn-based Heusler alloys have attracted much attention due to their unique properties and potential applications in many technological areas. One of the essential applications of Mn-based Heusler alloys is their use in the field of spintronics—the field of electronics where the transfer of energy and information is carried out not by an electric current but by a current of spins. Until now, it has been reported that quite a few Mn-based Heusler alloys are half-metals or spin-gapless semiconductors (SGS) [2–6]. Half-metallic materials exhibit high spin polarization reaching 100% near the Fermi level ($E_F$). These properties allow us to consider them as materials for magnetic sensors or non-volatile random-access memory devices [7]. Thus, their use in devices significantly reduces the energy loss [8].

The SGS are an intermediate state between the well-known half-metallic ferromagnets and gapless semiconductors. In SGS, one spin channel has an open bandgap near $E_F$, like a half-metal, and the other spin channel has a zero bandgap, like in a gapless semiconductor. Thus, conducting electrons or holes are not only 100% spin-polarized but can easily be transferred to an excited state. Among these Mn-based Heusler alloys, Mn$_2$CoZ (Z = Al, Ga) are of particular interest since they are not only theoretically predicted as half-metals/SGS but can also be realized experimentally [2,3,9–11].

Heusler alloys X$_2$YZ (X, Y are transition metals, Z is a chemical element from the main group) crystallize in a cubic structure of the Cu$_2$MnAl-type ($L2_1$ structure, space group $Fm\bar{3}m$), which has four interpenetrating *fcc* sublattices, namely A, B, C, and D with X, Y, and Z atoms occupying Wyckoff positions $8c$ (0.25.0.25.0.25), $4b$ (0.5.0.5.0.5), and $4a$ (0, 0, 0), respectively. The X atoms are equally occupied in the A and C sublattices. The Y and Z atoms occupy B and D sublattices, respectively. Apart from the usual Cu$_2$MnAl structure, Heusler alloys also crystallize in a structure of the Hg$_2$CuTi-type, commonly known as the inverse Heusler structure (XA structure, space group $F\bar{4}3m$). The main difference of this structure from the conventional $L2_1$ structure is the interchange between X atoms in the C sublattice and Y atoms in the B sublattice. These structural models are shown in Figure 1 for the case of X = Mn and Y = Co.

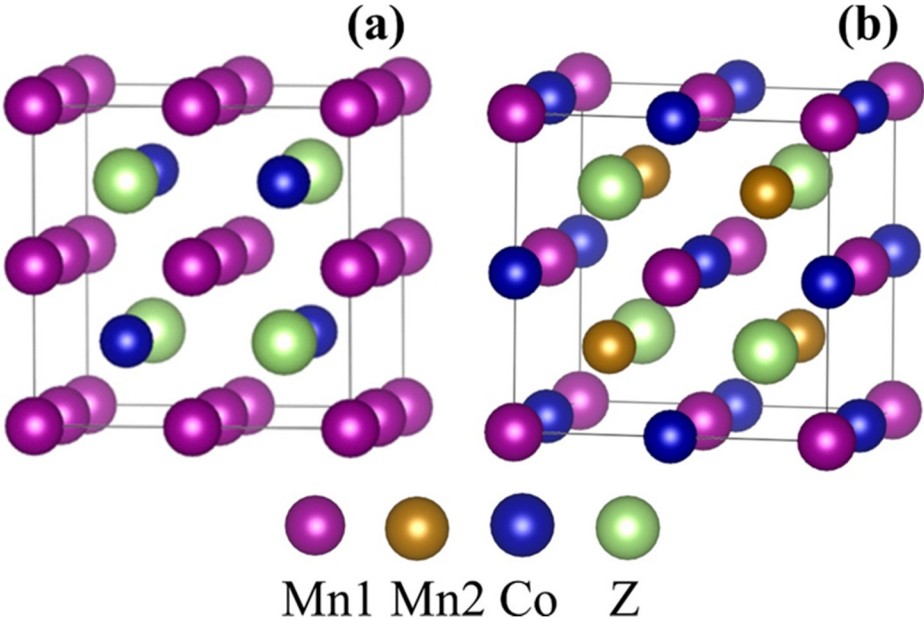

**Figure 1.** Unit cells of the Heusler alloys Mn$_2$CoZ of structure (**a**) $L2_1$ and (**b**) XA.

It was shown [12], that the Heusler alloy Mn$_2$CoAl has the electronic structure of SGS that is stable against tetragonal lattice distortions. However, it transforms into a semi-metallic state with an excess of Co or atomic swaps even between sites with different local symmetry. In the case of Mn$_2$CoAl, the doping decreases the bandgap while an increase in the bandgap is noticed for Mn$_2$CoGa. The half-metallicity is destroyed in both cases when the doping level is beyond a certain degree [12]. Despite numerous studies devoted to the electronic structure and magnetic properties of Heusler alloys, this problem remains not fully disclosed.

Recently, the existence of several fully compensated ferrimagnets in the Mn-Co-V-Al, Mn-Co-V-Si, and Mn-Co-V-Ga systems was predicted theoretically [13,14]. Some of the predicted half-metallic ferrimagnets Mn$_{2-x}$Co$_x$VAl were synthesized and studied [15,16]. The main drawback of these alloys is a low Curie temperature, which is an obstacle to the use of alloys in spintronics. For example, for the composition MnCoVAl, the Curie temperature $T_C$ is 105 K [17,18]. More suitable for the use as a spintronic material is Mn$_{1.5}$FeV$_{0.5}$Al with a Curie temperature $T_C = 355$ K and almost 100% spin polarization [19]. The highest

Curie temperatures were achieved for the Mn-Co-V-Ga system, exceeding 690 K for the $Mn_2V_{1-x}Co_xGa$ compositions (x = 0; 0.25; 0.5; 0.75; 1). The reported values of $T_C$ are the highest among the $Mn_2V$-based fully compensated ferrimagnets. Compensation of the magnetic moment without a significant decrease in $T_C$ indicates that the substitution of Co for V does not weaken the exchange interaction in the $Mn_2VGa$ alloys [19]. On the other hand, substitution of Co for Mn leads to a decrease in the Curie temperature in $Mn_{2-x}Co_xVGa$ alloys (x = 0.5; 0.75; 1) up to 171 K for MnCoVGa composition. In the mean-field approximation, it was found that the antiferromagnetic Mn-V exchange interaction is the strongest interaction and makes a larger contribution to the Curie temperature than ferromagnetic Mn-Mn and V-V exchange interaction [20]. In the $Mn_{2-x}Co_xVGa$ alloys, the Curie temperature correlates with the total magnetic moment. With an increase in cobalt concentration, the total magnetic moment per formula unit decreases, which is accompanied by a decrease in $T_C$. Consequently, the Curie temperature for $Co_2VGa$ is almost 300 K lower than that for $Mn_2VGa$. [21] Such a low magnetic ordering temperature of $Co_2VGa$ as compared to that of $Mn_2VGa$ is a consequence of a weak ferromagnetic interaction between Co and V atoms as compared to the strong antiferromagnetic interaction between Mn and V.

Combined study by neutron diffraction measurements and ab initio calculations revealed the crystal structure and magnetic configuration of the $Mn_2Co_{1-x}V_xZ$ (Z = Al, Ga) alloy. The neutron diffraction data and detailed ab initio studies confirmed the $L2_1$ structure for $Mn_2VGa$ and the XA structure for $Mn_2CoGa$. It was found that alloys with a high V concentration have the $L2_1$ structure. When the Co concentration reaches 75%, a transition from $L2_1$ to XA was observed [22]. When the content of Co in the substance is higher than 75%, the lattice constants change nonlinearly. With an equal Co and V atoms content, almost complete compensation of the magnetic moment (0.1 μB/f.u.) was found due to the ferrimagnetic coupling of Mn with V and Co [23].

Powder neutron diffraction experiments suggested that in Heusler $Mn_2CoGa$ the atomic arrangement can be described by the so-called $L2_{1b}$-type structure rather than by the XA structure [24]. The $L2_{1b}$-type structure is equivalent to the $L2_1$-type structure, where a partial disorder exists between Co and Mn atoms. In this case, the atomic configuration becomes "(Mn, Co), Mn, (Mn, Co), and Z." The magnetic moments obtained experimentally were in good agreement with the theoretical values in the model of the $L2_{1b}$ structure. At the same time, the calculations showed that the $L2_{1b}$ structure is metastable with respect to the XA structure [24].

In present work, we theoretically investigate the effect of substitution of Co for V on structural, magnetic, and electronic properties of $Mn_2Co_{1-x}V_xZ$ (Z = Al, Ga) Heusler alloy for the case of $L2_1$ and XA structures.

## 2. Materials and Methods

*Computation Details*

$Mn_2Co_{1-x}V_xZ$ (Z = Al, Ga) Heusler alloy for the case of $L2_1$, $L2_{1b}$, and XA structures density functional theory (DFT) with periodic boundary conditions using projector-augmented wave (PAW) [25] method and Perdew-Burke-Ernzerhof (PBE) functional [26] were performed using the ab initio total energy and molecular dynamics program VASP (Vienna Ab initio Simulation Package) [27–29]. All calculations were performed with the $4 \times 4 \times 4$ Monkhorst–Pack k-points sampling and 500 eV cut-off energy. The convergence tolerance for the calculations was chosen as the difference in total energy within $10^{-6}$ eV/atom. The following electronic configurations were used for the pseudopotentials: Mn $(3p^63d^54s^2)$, Co $(3d^74s^2)$, V $(3s^23p^63d^34s^2)$, Ga $(3s^23p^13d^{10})$, Al $(3s^23p^1)$, respectively. All calculations were performed using $2 \times 2 \times 2$ supercells of the XA and $L2_1$ structures, consisting of 128 atoms. Magnetic moments were computed using DDEC6 formalism [30,31].

## 3. Results and Discussion

**Structure**. In total, the unit cell of $X_2YZ$ contains four structural units. In this case, the XA symmetry cell contains two symmetrically nonequivalent atoms X (Mn), which we further denote as Mn1 (Mn2). These atoms differ in their nearest environment. Mn1 atoms are surrounded by 4 Mn2 and 4 Z atoms, while Mn2 atoms are neighbored by 4 Mn1 atoms and 4 Y atoms. Y atoms have 4 Z and 4 Mn1 atoms in the first coordination shell, and Z atoms are surrounded by 4Mn1 and 4Y atoms. This difference in the local environment determines the difference in the physical properties of the Mn1 and Mn2 atoms. In the $L2_1$ symmetry lattice, the X atoms are equivalent and are surrounded by 4 Y and 4 Z atoms.

We simulated the properties of $Mn_2Co_{1-x}V_xZ$ alloys when V atoms replace Co. Preliminary calculations have showed that when replacing Co atoms with V, it is energetically more favorable to arrange them at the most distant distances from each other. For this reason, the replacement of atoms was carried out randomly so that the average distance between the replacement atoms was maximized.

The interatomic distance maximization was achieved by arranging the V atoms so that, if possible, they did not occupy positions at the sites corresponding to the minimum distance between the pairs of Co-Co atoms at x = 0, equal to approximately 4A. Note that each Co atom is surrounded by 12 of the same type at the indicated distance. These characteristics are the same for both considered types of crystal lattices. When x is not equal to 0, the number of V-V pairs can be considered a characteristic of the local order in substituting Co atoms. In the case of uniform substitution of V for Co atoms, it can be expected that the number of pairs of V-V atoms will vary in proportion to their concentration. At x = 0.25, 0.5 and 0.75, there will be 3, 6, and 9 of such pairs, respectively. However, in the systems we studied, the number of these pairs of atoms was 1.5 (x = 0.25), 5.875 (x = 0.5), and 8.75 (x = 0.75). In this case, the average distance between all possible pairs of V-V atoms varied from approximately 6.4 Å (x = 0.25) to 5.9 Å (x = 1).

Comparison of the ground state energies $\Delta E = E(L2_1) - E(XA)$ of the considered crystalline modifications of the $Mn_2Co_{1-x}V_xZ$ compound (Figure 2) showed the presence of a phase transition near x = 0.5. At low x, the XA structure is more energetically favorable, while the $L2_1$ is more stable at high V concentrations. This result is in good agreement with the experimental data [22].

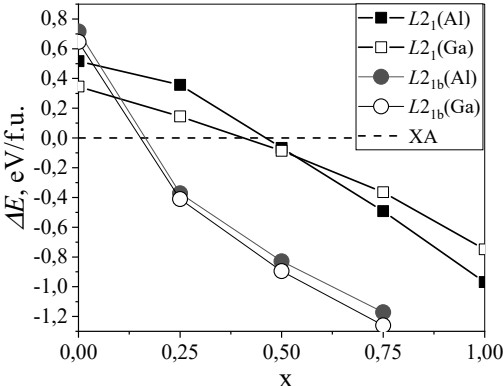

**Figure 2.** The ground state energies for the XA, $L2_1$ and $L2_{1b}$ structures in the $Mn_2Co_{1-x}V_xZ$ compositions as a function of V concentration x.

In addition to filling either B or C with the Co/V mixture, we also distribute V randomly on B and half of remaining Co atoms randomly on A and C sites (structure $L2_{1b}$). Calculations have shown that considering the $L2_1$ structure, the total energy of the systems decreases, and the $L2_1$ structure turns out to be more favorable. In the case of the XA structure, introducing such additional disorder leads to an increase in energy. As a result of comparing the energies of the $L2_{1b}$ and XA structures, the theoretical structural phase transition should occur at x = 0.25 (Figure 2), which is in excellent agreement with the experimental data [22].

The transition from one to another structure is explained by the difference in the energies of crystal formation ($E_{form}$) for modifications of the crystal structure. For the Heusler $X_2YZ$ alloys the $E_{form}$ values were calculated by the formula

$$E_{form}(X_2YZ) = E_{tot} - (2E(X) + E(Y) + E(Z)) \tag{1}$$

The data obtained for the end cases x = 0 and x = 1, of the $Mn_2Co_{1-x}V_xZ$ compounds are shown in Table 1. It can be seen that Co-containing alloys are more stable in the crystalline modification of the XA type, while the V-containing ones are more stable in the crystalline modification of the $L2_1$ type.

**Table 1.** Energies of crystal formation.

| Compound | $E_{form}$, eV/atom | | Reference |
| --- | --- | --- | --- |
| | **XA** | $L2_1$ | |
| $Mn_2CoGa$ | −0.211 | −0.127 | Present work |
| | −0.192 | | [11] |
| $Mn_2CoAl$ | −0.319 | −0.190 | Present work |
| | −0.286 | | [11] |
| $Mn_2VGa$ | −0.080 | −0.267 | Present work |
| $Mn_2VAl$ | −0.117 | −0.358 | Present work |

To explain this trend, we calculated an electron-localization function (ELF) to study the change in the nature of chemical bonds when replacing Co with V. The ELF is a ground-state property that is useful to visualize and distinguish between different bonding interactions in solids [32].

In regions with higher ELF values, the electrons tend to become paired, which indicates the covalent character of the bonding between the nearest-neighbor atoms in this intermetallic compound [33,34]. Figure 3 shows the ELF projections in the (110) plane for the Heusler alloys $Mn_2Co_{1-x}V_xZ$ (Z = Ga, Al) for x = 0 and x = 1 in the case of XA (Figure 3, left) and $L2_1$ (Figure 3, right) structures.

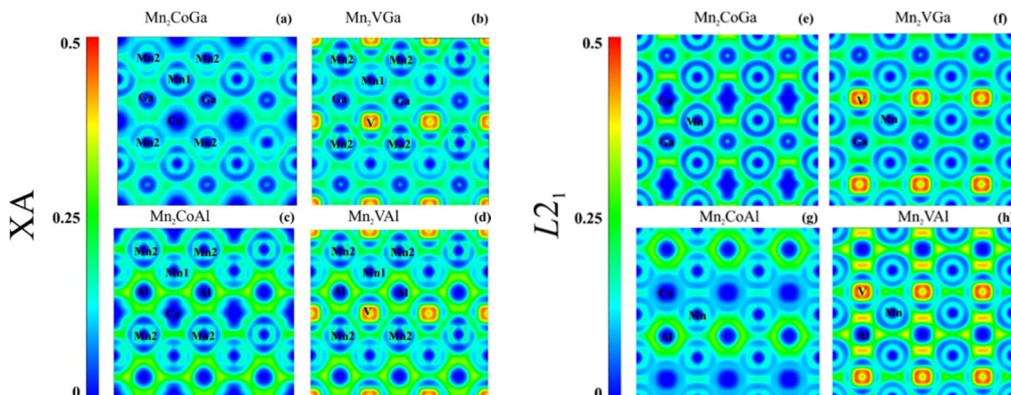

**Figure 3.** Calculated ELF in the (110) plane of (**a**) $Mn_2CoGa$, (**b**) $Mn_2VGa$, (**c**) $Mn_2CoAl$, and (**d**) $Mn_2VAl$ of XA and (**e**) $Mn_2CoGa$, (**f**) $Mn_2VGa$, (**g**) $Mn_2CoAl$, and (**h**) $Mn_2VAl$ of $L2_1$ structures.

Replacing Co with V leads to an increase in ELF, which means an increase in the covalent nature of the chemical bond between the elements (compare Figures 3a and 3b; Figures 3c and 3d; Figures 3e and 3f; Figures 3g and 3h). Strengthening the covalent bond should lead to an increase in the stability of the compounds. A competing effect leading to the destabilization of alloys is an increase in the degree of polarity of the covalent bond. An effect of this kind was found in the $Mn_2CoGa$ alloy when the Ga atoms were replaced by Ti, V, Cr, and Ni [34]. The relatively high ELF values around atoms serve as an indicator

of an increase in the polarity of the covalent bond [34]. In our case, high ELF values were found near V atoms. Thus, on the one hand, the substitution of V for Co atoms leads to an increase in the degree of covalence of chemical bonds in Heusler alloys, stabilizing the crystal lattices. On the other hand, the strong polarity of the covalent bond of V atoms with neighboring atoms leads to the destabilization of the crystal lattices. The crystal structures realized are the result of a compromise between these two trends.

The lattice parameters of the $L2_1$ and XA structures of the $Mn_2Co_{1-x}V_xZ$ compound behave differently depending on the V concentration (Figure 4).

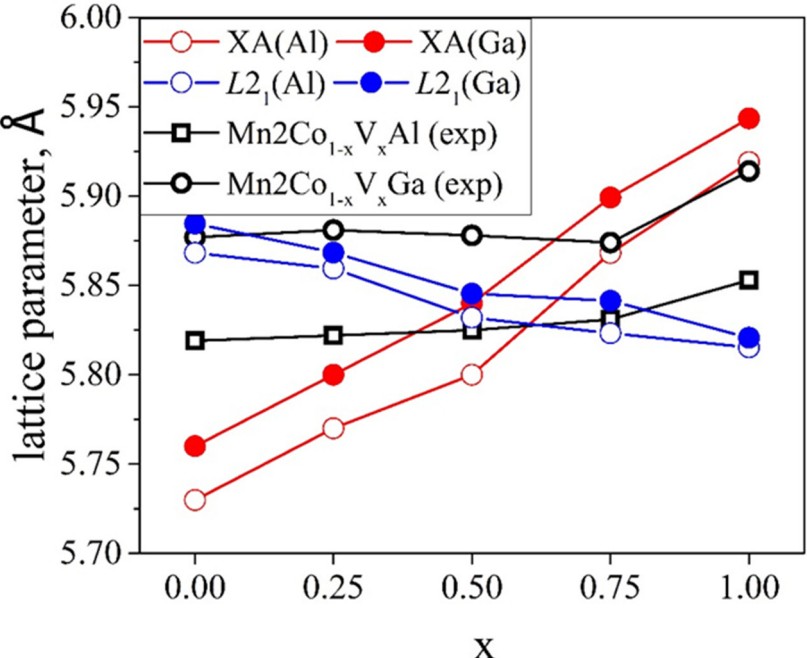

**Figure 4.** The calculated values of the lattice constants of the $Mn_2Co_{1-x}V_xZ$ (Z = Ga, Al) compounds for the case of the $L2_1$ and XA structures. Experimental data are adopted from Reference [20].

It can be seen that the XA structure is characterized by a linear increase in the lattice constants with increasing x. In the $L2_1$ structure, an increase in the V concentration leads to a slight decrease in the lattice parameters. A nonlinear dependence of the lattice constants was observed experimentally [20]. In experiment, the XA structure was detected at $x \geq 0.25$ [22]. The authors of [22] point out the difficulties in identifying the crystal structure associated with determining the arrangement of atoms in the lattices. In addition, the samples can represent a mixture of the $L2_1$ and XA structures, one of which can be metastable. From the point of view of theory, it is also necessary to take into account the possibility of considering not only the disorder at the Co-V sites but also the other types of disorder.

**Magnetic properties.** The results of calculating the average values of the magnetic moment on atoms of various types in the Heusler alloys $Mn_2Co_{1-x}V_xZ$ (Z = Ga, Al) for the case of the $L2_1$ and XA structures are shown in Figure 5. The magnetic moment values at x = 0, 0.25, and 1 are in good agreement with the known literature data.

In the case of the $L2_1$ ordering, as Co is replaced by V, the absolute values of the magnetic moments on the Mn and V atoms decrease, which is accompanied by an increase in the magnetic moments on the remaining Co atoms (Figure 5a,b). The values of the magnetic moments on the Mn and V atoms at x = 0 are almost 2 times larger than the corresponding values for x = 1. On the Mn and Co atoms, the direction of the magnetic moments is the same, while the moments on the V atoms have the opposite direction.

As compared with the $L2_1$ structure, the values of the magnetic moments on the atoms in the XA structure change more weakly with the increase in V content. The magnetic moments at the Mn1 atoms undergo the most remarkable change. The absolute value of

the magnetic moment increases at x = 1 by approximately 0.8 μB in the x = 1 composition as compared to the corresponding value in the x = 0 composition. In this case, the magnetic moments on the Mn1 and V atoms are parallel, and the moments on the Mn2 and Co atoms are antiparallel (Figure 5c,d).

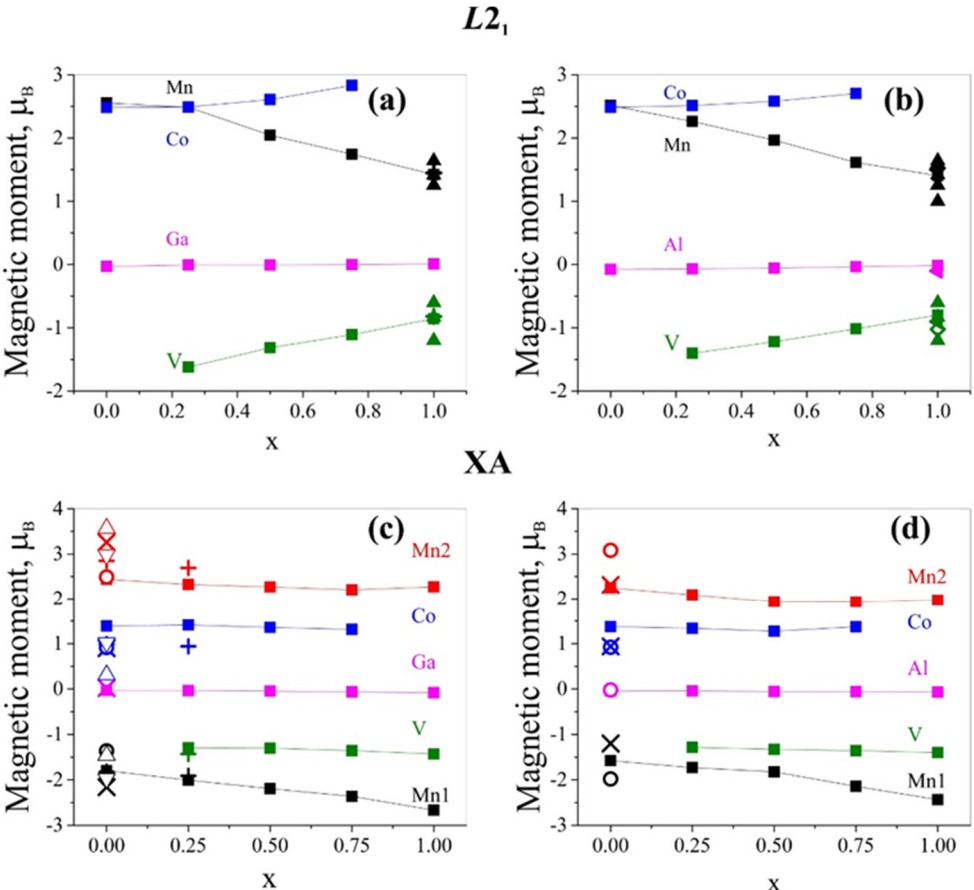

**Figure 5.** Average values of magnetic moment on atoms calculated by the DDEC6 method on atoms of the Heusler alloy $Mn_2Co_{1-x}V_xZ$ of the $L2_1$ structure (**a**) Z = Ga, (**b**) Z = Al and XA structure (**c**) Z = Ga, (**d**) Z = Al. Literature data are given for comparison: ×—Ref. [2], ○—Ref. [13], +—Ref. [22], △—Ref. [24] (theory), ▽—Ref. [24] (experiment). ◄ [2], • [35], ◊—[36], ▲—[37].

The presence of two types of Mn atoms in the XA-type lattice, which have the opposite direction of the magnetic moment, determines the difference in the magnetic properties of these crystalline modifications. The calculated values of the total magnetic moment of the XA and $L2_1$ structures are shown in Table 2.

The most significant changes in the values of the magnetic moments are observed for Mn and V atoms in the case of Heusler alloys with the $L2_1$ structure. A decrease in the absolute values of the magnetic moments of atoms occurs here with an increase in the concentration of V atoms in the first coordination sphere of Mn atoms. For the Heusler alloy of the XA structure, the Mn1 and V atoms are located at the sites of the second coordination sphere of each other. In this case, the increase in the absolute value of the magnetic moment at the Mn1 atoms is determined by the number of V atoms at the next-nearest neighbors Mn1 positions.

The value of the total magnetic moment in the XA ordering obeys the Slater-Pauling rule (S-P rule) [35]. According to this rule, the total magnetic moment in half-metallic Heusler alloys can be found by the formula $M_t = Z_t - 24$, where $Z_t$ is the total number of valence electrons per unit cell, and $M_t$ is the total magnetic moment per formula unit. The value of the total magnetic moment of the $L2_1$-type lattices does not obey this rule due to the ferromagnetic coupling of the Mn and Co atoms, except the case x = 1. The decrease in

the value of the total magnetic moment with increasing x occurs due to the contribution of the antiferromagnetically coupled spins of the V atoms. In contrast to the $L2_1$ structure, the spins of Mn1-V and Mn2-Co atom pairs in the XA structure have the opposite direction. In order to make the $Mn_2Co_{1-x}V_xZ$ compositions obeying the S-P rule in the $L2_1$ structure, it is necessary to take into account the disordering of pairs of Mn-Co atoms, i.e., to consider the structure $L2_1$ [24,38]. Taking this kind of disorder into account is beyond the scope of this work.

**Table 2.** Total magnetic moment calculated for the $L2_1$ and XA structures of the $Mn_2Co_{1-x}V_xZ$ (Z = Al, Ga) Heusler alloy.

| Z | x | Total Magnetic Moment Per Formula Unit, $\mu_B$/f.u. | |
| --- | --- | --- | --- |
| | | XA Structure | $L2_1$ Structure |
| Ga | 0 | 2.00 [a], 2.05 [b], 2 [c], 1.99 [d] | 7.71 [a], 7.68 [d] |
| | 0.25 | 1.00 [a], 1.11 [b], 1 [c] | 6.43 [a] |
| | 0.5 | 0.06 [a], 0.1 [b], 0 [c] | 4.73 [a] |
| | 0.75 | 1.05 [a], 0.97 [b], 1 [c] | 3.36 [a] |
| | 1 | 2.01 [a], 1.8 [b], 2 [c] | 1.98 [a] |
| Al | 0 | 2.00 [a], 2.06 [b], 2 [c] | 7.44 [a] |
| | 0.25 | 1.00 [a], 1.09 [b], 1 [c] | 5.99 [a] |
| | 0.5 | 0.06 [a], 0.06 [b], 0 [c] | 4.56 [a] |
| | 0.75 | 0.93 [a], 0.99 [b], 1 [c] | 3.11 [a] |
| | 1 | 1.94 [a], 1.863 [b], 2 [c] | 2.00 [a] |

[a]—Present work, [b]—Experimental data from Ref. [12], [c]—S-P rule, [d]—Theoretical data from Ref. [24].

**Electronic properties.** Figure 6 shows the calculated electronic densities of states for the Heusler alloys $Mn_2Co_{1-x}V_xZ$ (Z = Ga, Al) of the XA structure for various x values.

Since our theoretical consideration indicated that in the $L2_1$ type ordering there is no compensation of the magnetic moment required for practical applications, we did not investigate the dependence of the electronic structure of these compounds for the case of $L2_1$ ordering. The electronic structure of $Mn_2VAl$ and $Mn_2VGa$ for the case of $L2_1$ ordering was studied in References [2,38], respectively.

According to our calculations (Figure 6), $Mn_2CoGa$ is a half-metal, while $Mn_2CoAl$ is SGS, which is consistent with previous studies [10–12,39]. As in the case of replacement of the Z element by Fe or Cr [12], the replacement of Co by V destroys the SGS state of $Mn_2CoAl$ and its electronic structure becomes half-metallic. As x increases to 0.5, the density of states of spin-up electrons near the Fermi level increases and is formed due to the electronic states on Mn1, Mn2, and V atoms. In this case, the fraction of states near $E_F$ on the V atoms increases.

In the $Mn_2CoGa$ compound, the replacement of Co by V leads to a change in the electronic structure towards an enhancement of the half-metallic properties. However, the obtained electronic structure does not fully correspond to the standard definition of half-metal, where there must be an energy gap at the Fermi level for spin-minority states. Instead, spin-minority states exhibit properties characteristic of semi-metals rather. As a result, the electronic DOS, in this case, is characterized by a high density of states for spin-majority states and a low density of states (by about two orders of magnitude) for spin-minority states. For this reason, we will further call compounds with such an electronic structure as semi-half-metals. As x grows, the DOS minimum at x = 0 near the Fermi level is gradually replaced by a peak. DOS near the Fermi level for the spin-up states, as in the case of $Mn_2CoAl$, is formed by the states on Mn1, Mn2, and V atoms with an increase in the fraction of states on V atoms as x increases. In all the cases considered, the energy gap for spin-down states when Co atoms are replaced by V is filled with d-states on Mn1 and V.

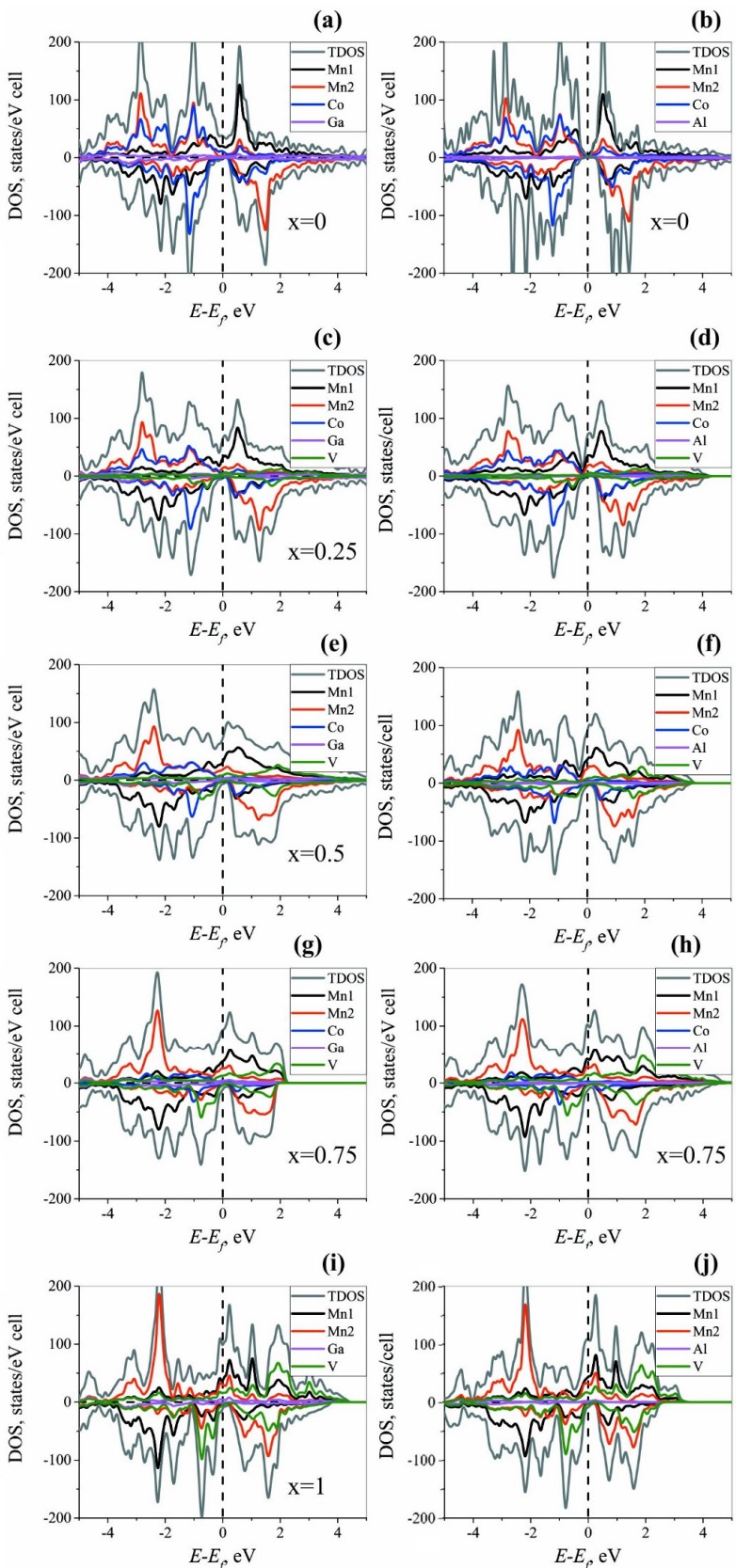

**Figure 6.** Electronic DOS of the Mn2Co1-xVxZ (Z = Ga, Al) Heusler alloys for the case of the XA structure. Panels (**a,c,e,g,i**) correspond to $Mn_2Co_{1-x}V_xGa$ alloys at x = 0, 0.25, 0.5, 0.75 and 1, respectively. Panels (**b,d,f,h,j**) correspond to $Mn_2Co_{1-x}VAl$ alloys at x = 0, 0.25, 0.5, 0.75 and 1, respectively.

In order to examine details of the electronic structure near the Fermi level, one has to examine the band structure around the Ef since band crossing is not always easily distinguished in usual DOS calculations. The results obtained for ordered alloys of the XA structure are shown in Figure 7. For the $Mn_2CoAl$ alloy, the data obtained are in complete agreement with the results of [12]. The rest of the compounds exhibit semi-half-metallic behavior. In this case, the energy levels for spin-down states intersect the Fermi level, which indicates that their DOS near Ef is not precisely equal to zero.

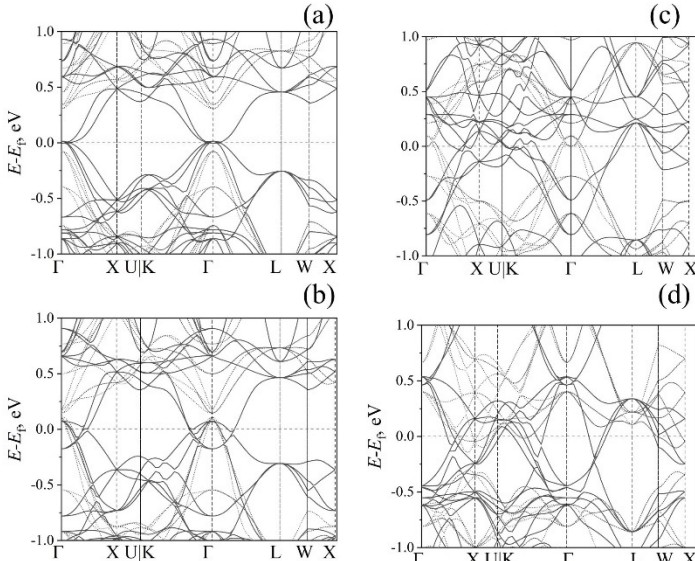

**Figure 7.** Spin-resolved band structures of ordered (**a**) $Mn_2CoAl$, (**b**) $Mn_2CoGa$, (**c**) $Mn_2VAl$, (**d**) $Mn_2VGa$ Heusler alloys for the case of the XA structure (solid lines represent spin-up states; dotted lines correspond to spin-down states).

## 4. Conclusions

This paper presents the results of DFT studies of the structural, electronic, and magnetic properties of Heusler alloys $Mn_2Co_{1-x}V_xZ$ (Z = Ga, Al, x = 0, 0.25, 0.5, 0.75, 1) for the case of $L2_1$ and XA structures. It is shown that for x = 0, the XA structure is more stable, while for x = 1 the $L2_1$ structure is stable. A transition from one to another type of ordering occurs near x = 0.5. The inclusion of additional atomic disorder through consideration of the lattices of the $L2_{1b}$ structure improves the agreement between theory and experiment, and the phase transition occurs at a vanadium concentration x = 0.25. Depending on the degree of Co substitution for V, the lattice constants of the $L2_1$ and XA structures change in different ways, explaining the experimentally observed trends of the lattice constants. Almost complete compensation of the magnetic moment was achieved in the $Mn_2Co_{0.5}V_{0.5}Z$ (Z = Al, Ga) composition for the case of the XA structure. The compensated magnetic moment for these alloys is 0.06 $\mu_B$/f.u. The calculated values of the magnetic moments of the alloys with XA ordering correspond to the values calculated according to the S-P rule. In order to achieve agreement with the S-P rule for the $L2_1$ structure, it is necessary to take into consideration other types of the atomic disorder, such as swap between Co and Mn atoms. Substitution of V for Co atoms leads to a change in the electronic structure of the alloy from SGS to a semi-half-metallic state. At the same time, the bandgap for spin-minority states disappears, and a pseudogap state is formed with DOS 2 orders of magnitude smaller than the same value for spin-majority states.

**Author Contributions:** Author contributions F.A. and T.I. conceived the study, designed the theoretical calculations and wrote the manuscript. A.A. performed most of the calculations. N.M. performed the theoretical calculations. N.S. and G.K. analyzed simulation data. M.S. and V.K. contributed to DFT simulations. All authors contributed to manuscript revision. All authors have read and agreed to the published version of the manuscript.

**Funding:** This work was carried out within the framework of grant from the Ministry of Education and Science of the Republic of Kazakhstan (project AP08957176 "First-principles design of fully compensated ferrimagnetic materials for spintronic applications").

**Institutional Review Board Statement:** Not applicable.

**Informed Consent Statement:** Not applicable.

**Data Availability Statement:** Not applicable.

**Acknowledgments:** The work of T.I. was performed under the state assignment of IGM SB RAS. T.I. also thanks the Center for Computational Materials Science (IMR, Tohoku University) for access to the supercomputing system to perform the simulations. Th calculations were partially performed at the Cherry supercomputer cluster provided by the Materials Modeling and Development Laboratory at NUST "MISIS" (supported via the Grant from the Ministry of Education and Science of the Russian Federation No. 14.Y26.31.0005).

**Conflicts of Interest:** There are no conflicts to declare.

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
