# Peer review of "Structural, Electronic and Magnetic Properties of Mn2Co1-xVxZ (Z = Ga, Al) Heusler Alloys: An Insight from DFT Study"

_magnetochemistry, doi:10.3390/magnetochemistry7120159_

Round 1
Reviewer 1 Report
This is a comparably nice DFT work on the structure, magnetic and electronic configurations of new Heusler systems. In contrast to many other comparable works published by journals in this league, it is not an arbitrary theoretical investigation of some system that perhaps can never be synthesized due to instability, but was done in response to previous experimental work.
I did not find any major problems with the submission -- there are a few points I will list below. Thus, in principle it could be accepted as it is. However, I do have one main point that the authors could consider, and I would at least like to hear their opinion on it:
From the point of view of solid state chemistry, the behaviour you report here is quite expected -- in Heusler systems (with Cu2MnAl as prototype) it is normally a main group element on sublattice D (using your convention), a late transition metal on A and C, and an early transition metal on B. You substitute now Co for V in the role of element Y. V is earlier in the periodic table than Mn, so it will readily go on B, leaving Mn on A and C, corresponding to conventional L2_1 order. On the other hand, Co is later than Mn, so it will rather go itself on A or C and push Mn onto B. However, my point is now that the inverse Heusler structure is only the obvious solution if you have exclusively Co. But what will happen for 0<x<1? From this point of view, I think the most obvious solution is to have then B partially occupied by V, A and C partially by Co, and Mn filling the rest. This would be L2_1-symmetry. At low temperatures, the system would in principle have an ordered ground-state with some long-range arrangement of Co and V on their respective sublattices, but it is well possible that for small Co contents the ordering energies are too small and thus the ordering temperatures too low to experimentally allow ordering on accessible timescales. Only for large Co contents, ordering energies become high, so that the ordered state is reachable, which in this case is the inverse Heusler structure. I really think that this is what they saw in the experiment.
How this pertains to your investigation: I would really propose that you try, in addition to filling either B or C with the Co/V mixture, to distribute V randomly on B and Co randomly on A and C. The computational effort would be exactly the same, so this is doable ("outside the scope of the study" won't be a good excuse), and I would expect the result to be even lower in energy.
To the minor points:
On page 4, Mn1 has 4 Mn2 and 4 Z as neighbours in the inverse Heusler structure. Y are only the next-nearest neighbours, and there are 6 of them. Also, Mn2 has 4 Mn1 and 4 Y as next neighbours, Z are next-nearest neighbours, and of course also in this case there are 6 of them.
You write "the replacement of atoms was carried out randomly so that the average distance between the replacement atoms was maximized" -- if it was really random, then I do not see which measure of "average distance" would be maximized. Or was it some special random configuration?
On page 6, you mistakenly write that for large V concentrations theory predicts the inverse Heusler. This is incorrect, both you and [22] predict L2_1 in this case (note that their x is defined just the other way around than yours, is this deliberate?). Thus, there is no contradiction in this sense. But yes, according to your calculations, the lattice constant should be concave as function of x, while experimentally it is convex. Perhaps this is because the correct structure for 0<x<1 is like I sketched above?
Finally, due to the necessary smoothing in the DOS in Fig. 6 I cannot decide whether the density at the Fermi energy is really zero. In this case, the values in Table 2 should be integer (the deviations are probably due to occupation smearing, or not?). Band structures (at least for the ordered end-members, where they are easy to calculate) would help here.
Author Response
Please see the file attached.

Reviewer 2 Report
The authors reported a systematic investigation on the structural, slectronic, and magnetic properties of Mn2Co1-xVxZ (Z = Ga, Al) Heusler Alloys by first principles calculation. The obtained results are interesting and the whole paper is designed properly. I have some minor commomts before it is accpeted, as follow: 1) Since all calculations are performed with 222 supercell, different magnetic configurations can and should also be considered in addition to the simple one as discussed in the manuscript, such as different magnetic orientation. 2) Some minor language problem and formatting errors should be carefully revised. 3) How the different compositions with V concentration x are obtained, the authors should give clear description.Author Response
Please see the file attached

Reviewer 3 Report
This paper describes a DFT examination of some Heusler alloys that are experimentally known, allowing for a comparison to data. The authors are trying to replicate the physical properties (structural, magnetic, electrornic) of the known materials and are moderately successful with this approach. The structures are reproduced and qualitatively evolve with composition as expected. Density of states calculations are consistent with experimental properties. Moments on individual ions qualitatively match experimental data. Known disorder in the experimental materials complicates the analysis. I don't find fault with the experimental approach or the conclusions. My only concern about this paper is that it while it tries to fit the experimentally known data, it doesn't seem to offer much insight into the design of new materials, which would make the contribution so much more impactful. Even incremental suggestions for new compositions, (even incorrect suggestions) would render this approach of using computation as a guide more legitimate. Nevertheless, I can see how this work complements the work of others. The paper needs some editing and I have attached some of the suggestions I have
Author Response
Please see the file attached.

Round 2
Reviewer 1 Report
Your results show that L2_1b, putting V on B and Co randomly on A and C, with Mn making up the rest, is always the energetically best solution, apart from pure Mn2CoZ (where the system can gain energy by concentrating all Co on either A or C) -- for pure Mn2VZ it is of course equivalent to your previous L2_1 model. I really think that you should not defer this to another paper, but treat it already here, as you now know that what you have calculated here is not the ground state.
And I would not say that the fit with experiment is worse: according to [22], the transition between L2_1 and X_A is somewhere between x=0.25 and x=0.5 (in your convention). If you consider only L2_1, you get somewhat around x=0.45, with L2_1b you get x=0.15. And remember that in [22] they didn't consider L2_1b when fitting their occupations, so this is not so reliable information.
Why didn't you respond to my point about the number of neighbours of the different sites in X_A? As it is, your description on page 4 is still incorrect.
And also for the supposed discrepancy between the x where your theoretical results and the experiments of [22] predict X_A: now you have it doubly incorrect -- [22] says X_A for high Co content, meaning x<=0.25 in your convention, and also you say X_A for x<=0.25. And no, this was not another experiment, but the same experimental data were in [20] evaluated only in terms of the L2_1 structure, while in [22] they recognized that X_A sometimes fitted better. So you should delete the sentence "There, the crystal structure was identified rather as L2_1", drop the "In another experiment", report only the results of [22], and also drop the sentence "Thus, although..." together with the next sentence "This contradiction...". There is no contradiction if you would just read these publications correctly!
And finally: the band structures in Fig. 7 show that apart from Mn2CoAl all other compounds have some density at the Fermi energy in both spin channels, thus the preceding paragraphs are not correct. It is not much, admittedly, but these are not half-metals (half-metals have a gap at E_F). Perhaps you want to call them semi-halfmetals, in analogy to semimetals, which have a small non-polarized electronic density of states at E_F. And in Fig. 7c you seem to have swapped spin up and down, isn't it?
Author Response
File with response is attached

Round 3
Reviewer 1 Report
If we are talking about nearest neighbours, then the situation in XA is actually quite simple: both Mn1 and Y have 4 Mn2 and 4 Z atoms as neighbours, and the other way around: both Mn2 and Z have 4 Mn1 and 4 Y as neighbours. Thus, the sentence in lines 139--141 "In the XA lattice, the Y atoms are surrounded by 4 Mn2 atoms and 4 Z atoms, while the Z atoms are surrounded by 4 atoms Mn1 and 4 atoms Y." is correct, but it repeats information that was already given (incorrectly) five lines above -- I would suggest you delete the sentence I quote above and correct the "Mn1" in line 135 to "Mn2", or perhaps better if you simply state it as I have done, namely by explicitly using the fact that Mn1 and Y have the same neighbours (thus you need to specify these neighbours only once), and the same for Mn2 and Z.
Further, please mention also in the abstract the L2_1b structure: you have calculated XA, L2_1 and L2_1b, found that XA is favourable compared to L2_1 at low V concentrations, but that L2_1b is always better apart from pure Mn_2CoZ, which means that these systems can be understood by assuming that Co goes on the sublattices where in conventional Heuslers the late transition metal sits, V goes on the sublattice where the early transition metal sits, and Mn (being between those two) makes up the rest. To me, this is the essential finding of your study, and it should be stated in the abstract.
If those two changes are made, I can recommend publication of the manuscript.
Author Response
File with our answers is attached
